# Investigating the Role of Micromammals in the Ecology of *Coxiella burnetii* in Spain

**DOI:** 10.3390/ani11030654

**Published:** 2021-03-02

**Authors:** David González-Barrio, Isabel Jado, Javier Viñuela, Jesús T. García, Pedro P. Olea, Fernando Arce, Francisco Ruiz-Fons

**Affiliations:** 1Instituto de Investigación en Recursos Cinegéticos IREC (CSIC-UCLM-JCCM), Ronda de Toledo 12, 13071 Ciudad Real, Spain; Javier.Vinuela@uclm.es (J.V.); jesusgarcia.irec@gmail.com (J.T.G.); 2Parasitology Reference and Research Laboratory, Spanish National Centre for Microbiology, Health Institute Carlos III, Ctra. Majadahonda-Pozuelo Km 2, Majadahonda, 28220 Madrid, Spain; 3Viral Hepatitis Reference and Research Laboratory, Spanish National Centre for Microbiology, Health Institute Carlos III, Ctra. Majadahonda-Pozuelo Km 2, Majadahonda, 28220 Madrid, Spain; 4Special Pathogens Reference and Research Laboratory, Spanish National Centre for Microbiology, Health Institute Carlos III, Ctra. Majadahonda-Pozuelo Km 2, Majadahonda, 28220 Madrid, Spain; ijado@isciii.es; 5Departamento de Ecología, Universidad Autónoma de Madrid (UAM), 28049 Madrid, Spain; pedrop.olea@uam.es; 6Centro de Investigación en Biodiversidad y Cambio Global (CIBC-UAM), Universidad Autónoma de Madrid, 28049 Madrid, Spain; 7School of Natural Sciences, University of Tasmania, Hobart, TAS 7004, Australia; fernando.arcegonzalez@utas.edu.au

**Keywords:** micromammals, *Coxiella burnetii*, Q fever, zoonosis

## Abstract

**Simple Summary:**

*Coxiella burnetii*, the causal agent of human Q fever and animal Coxiellosis, is a zoonotic infectious bacterium with a complex ecology that replicates in multiple host species. However, the role of wildlife in its transmission is poorly understood. We examined 816 spleen samples obtained from ten species of micromammals and 130 vaginal swabs from *Microtus arvalis* females to detect the presence of *C. burnetii* DNA by qPCR. Our aim was assessing whether infection occurs in micromammals in Spain and what species could be relevant hosts in pathogen maintenance. The 9.7% of the spleen samples were qPCR positive. The infection prevalence level was highest (10.8%) in *Microtus arvalis* and also one vaginal swab was PCR positive. Positive samples were also found in *Apodemus sylvaticus* (8.7%), *Crocidura russula* (7.7%), and *Rattus rattus* (6.4%). A genotype II+ strain was identified in one of the positive samples from *M. arvalis*. The results of the study are consistent with previous findings suggesting susceptibility of micromammals to *C. burnetii* infection. We also provide further support to consider micromammals when tracing the origin of human Q fever cases in Europe as one of the authors probably got infected while handling *M. arvalis*.

**Abstract:**

*Coxiella burnetii*, the causal agent of human Q fever and animal Coxiellosis, is a zoonotic infectious bacterium with a complex ecology that results from its ability to replicate in multiple (in)vertebrate host species. Spain notifies the highest number of Q fever cases to the ECDC annually and wildlife plays a relevant role in *C. burnetii* ecology in the country. However, the whole picture of *C. burnetii* hosts is incomplete, so this study seeks to better understand the role of micromammals in *C. burnetii* ecology in the country. Spleen samples from 816 micromammals of 10 species and 130 vaginal swabs from *Microtus arvalis* were analysed by qPCR to detect *C. burnetii* infection and shedding, respectively. The 9.7% of the spleen samples were qPCR positive. The highest infection prevalence (10.8%) was found in *Microtus arvalis*, in which *C. burnetii* DNA was also detected in 1 of the 130 vaginal swabs (0.8%) analysed. Positive samples were also found in *Apodemus sylvaticus* (8.7%), *Crocidura russula* (7.7%) and *Rattus rattus* (6.4%). Positive samples were genotyped by coupling PCR with reverse line blotting and a genotype II+ strain was identified for the first time in one of the positive samples from *M. arvalis*, whereas only partial results could be obtained for the rest of the samples. Acute Q fever was diagnosed in one of the researchers that participated in the study, and it was presumably linked to *M. arvalis* handling. The results of the study are consistent with previous findings suggesting that micromammals can be infected by *C. burnetii*. Our findings additionally suggest that micromammals may be potential sources to trace back the origin of human Q fever and animal Coxiellosis cases in Europe.

## 1. Introduction

*Coxiella burnetii* is a multi-host bacterium that causes Q fever in humans, a zoonosis that is emerging worldwide [1]. In humans, Q fever is associated with a multiple clinical spectrum, from asymptomatic to fatal disease. A low percentage of acute cases, especially patients with previous valvulopathy and, to a lesser extent, immunocompromised persons and pregnant women, develop chronic disease that may present with endocarditis, vascular alterations, chronic hepatitis, chronic pulmonary infections, or the so-called post-Q fever fatigue syndrome [2].

It is assumed that domestic ruminants are the main reservoir of *C. burnetii* for humans. Nonetheless, the origin of several human Q fever cases remains unclarified [3] and human–wildlife interaction has been suggested as a risk factor for human infection with *C. burnetii* [4]. The current changes in the patterns of wildlife–human interactions caused by variations in human and wildlife population dynamics and behaviour imply an increased risk of *C. burnetii* inter-species transmission [4]. The ecology of *C. burnetii* in wildlife is still poorly understood and the influence of host, environmental and pathogen factors is almost unknown [4]. *C. burnetii* infection has been neglected in wildlife despite the evidence of particular wild species behaving as true *C. burnetii* reservoirs [4]. Indeed, scientific publications focused on *C. burnetii* in livestock outnumber those in wildlife tenfold. The circulation of *C. burnetii* in wild vertebrates in the sylvatic cycle may perhaps be also enhanced by tick-borne transmission [5]. Wild terrestrial small mammals such as rodents are thought to constitute maintenance hosts of infection in the domestic cycle of *C. burnetii* [6,7,8,9]. 

Within Europe, Spain has reported the highest number of human Q fever cases annually since 2016 (Q fever is of mandatory notification in Spain since 2015). In 2018, Spain accounted for more than a third of the overall number of cases with 418 reported notifications [10]. Livestock (mainly cattle, sheep and goats) is an important reservoir of *C. burnetii* for humans in Spain [11]. However, the geographical location of the country (between Mediterranean and Atlantic oceanic climates) and its orography account for a wide diversity of habitats and biotopes that make Spain a European biodiversity hotspot. As *C. burnetii* is a multi-host pathogen by evolution, the implication of wild reservoirs in its life cycle was expected and already reported in different studies [12,13,14]. The role of wild micromammals such as rodents and insectivores in *C. burnetii* ecology is currently poorly known. A recent review of studies performed in wild mammals suggests that several micromammal species worldwide may be relevant hosts for *C. burnetii* [4]. It would be expected that the large diversity of micromammals in Spain would add up with joint effects favouring the proliferation of *C. burnetii* because of the higher host availability. Furthermore, the current expansion of some ‘pest’ rodent species (e.g., *Microtus arvalis*) would have an impact in the sylvatic cycle of the bacterium and therefore in the incidence of human Q fever and animal coxiellosis in the country. A large area in Spain is occupied by farming areas that are in line with the intensification trend of the agriculture in Europe, and it has suffered tremendous transformations in the last 2–3 decades [15]. This transformation is behind the massive spatial expansion and the cyclic population outbreaks of the common vole (*M. arvalis*) that is considered a severe agricultural pest at European level, a matter of intersectoral conflict and a risk for human and animal health [15,16]. Furthermore, the sustained human migration from rural to urban areas over the last four decades in Spain has notably contributed to re-wilding of Iberian forests [17], consequently bearing a re-colonization of lost areas by wildlife, including wild forest micromammals. In both agricultural and natural (forested) landscapes in Spain, the (direct and indirect) interaction with humans, grazing livestock and other wildlife may constitute a risk factor for the exchange of specific strains of *C. burnetii* among different hosts.

According to these premises and to the hypothesis of a relevant implication of wild micromammals in *C. burnetii* ecology in Spain (and beyond), the objectives of this study were to estimate the presence and prevalence of the bacterium in different wild micromammal species and phylogenetically characterize the *C. burnetii* genotypes present in these animals as a first stage to estimate the implication of wild micromammals in the epidemiology of Q fever.

## 2. Materials and Methods

### 2.1. Sampling

Between 2003 and 2014, and in the framework of different studies, samples from wild micromammals were collected in 16 locations in mainland Spain using LFATDG Sherman Live Traps (7.62 cm × 8.89 cm × 22.86 cm, H. B. Sherman Traps, Inc., Tallahassee, FL, USA) (Figure 1 and Table 1). Capture and handling procedures for sampling were approved by the UCLM Ethics Committee (reference number CEEA: PR20170201) and were in accordance with the Spanish and European policy for animal protection and experimentation. The researchers and technicians involved in the captures only employed gloves as personal protective equipment. Some of the individuals captured were randomly selected, sedated with an intramuscular injection of a solution containing Ketamin (10 mg/kg) and medetomidine (1 mg/kg) and thereafter humanely euthanised by cervical dislocation. These animals were transported refrigerated to our labs where a detailed necropsy was performed under biosafety 2 containment in cabinets, and tissue samples were collected and preserved frozen at −20 °C. Vaginal swabs (Aluminium + viscose AMIES swabs, Deltalab, Spain) were collected from live female *M. arvalis* captured in northwestern Spain along 2012. The swabs were thereafter preserved frozen at −20 °C until DNA purification. Some species (*Arvicola terrestris*, *Sciurus vulgaris* and *Eliomys quercinus*; Table 2) were surveyed after being found dead close to trap capture sites or by environment agents and brought to the lab for necropsy.

### 2.2. DNA Extraction, Coxiella burnetii DNA Detection and Genotyping

Spleen samples were the target tissue to estimate the occurrence of infection with *C. burnetii* because of the presence of *C. burnetii* DNA in this organ could only be the consequence of a generalized infection. DNA from spleen samples and swabs was extracted by using the DNeasy Blood and Tissue Kit (QIAGEN, Hilden, Germany). Around 25 mg of spleen from each animal was cut into small pieces on a sterile glass plate with a disposable scalpel blade before being disrupted in 180 µL of ATL buffer with a homogenizer (TissueLyser II, QIAGEN, Hilden, Germany). After disruption, samples were incubated at 56 °C for 1 h with 20 µL of Proteinase K. Later on, samples were vortexed for 15 s and, after adding 200 μL of AL buffer, the manufacturer’s blood extraction protocol was followed (http://mvz.berkeley.edu/egl/inserts/DNeasy_Blood_&_Tissue_Handbook.pdf accessed on 14 October 2020). The swabs were incubated at 56 °C for 30 min in 200 μL of AL buffer containing 20 μL of proteinase K in sterile 1.5 mL nuclease-free tubes. Swabs were then vortexed for 15 s and carefully removed after squeezing out the liquid contained in them with a sterile glass rod into the tube. The remaining solution was incubated at 56 °C for 30 min. The manufacturer’s blood extraction protocol was thereafter used. The DNA concentration in aliquots was quantified (NanoDrop 2000c/2000 spectrophotometer; Thermo Scientific, Waltham, MA, USA) and, if above 50 ng/µL, they were homogenised to that concentration with RNase/DNase free water (Promega, Madison, WI, USA). DNA aliquots were preserved frozen at −20 °C until the PCR was performed. Sample cross-contamination during DNA extraction was excluded by including negative controls (nuclease-free water; Promega, Madison, WI, USA) every ten samples that were also tested by PCR. A screening assay was selected for the detection of *C. burnetii* DNA in samples based on the IS1111-based PCR. Positive samples were further analysed by coupling the PCR with hybridization with a specific probe by reverse line blotting (RLB) [19,20]. The resulting genotypes were further analyzed with InfoQuest™FP 4.50 (BioRad, Hercules, CA, USA). Cluster analyses used the binary coefficient (Jaccard) and UPGMA (Unweigthed Pair Group Method Using Arithmetic Averages) to infer the phylogenetic relationships (Appendix A).

## 3. Results and Discussion

We detected *C. burnetii* DNA—positive if qPCR cycle threshold (Ct) values were <40.0—in 79 of 816 spleen samples analysed (Table 2). In general, and considering overall all locations, *Microtus arvalis* was the species displaying the highest ratio of infection with *C. burnetii* (10.8%; 62/572) followed by *A. sylvaticus* (8.7%; 12/138), *Crocidura russula* (7.7%; 2/26) and *R. rattus* (6.4%; 3/47). The other micromammal species sampled were negative. One of the 130 vaginal swabs collected from *M. arvalis* females was qPCR positive (0.8%; 95%CI: 0.1–4.2). Ten positive samples with Ct values < 35.0 were analysed by RLB hybridization. Only a genotype II+ strain could be obtained in one *M. arvalis* from Northwest Spain. The acute disease antigen A (*adaA*) gene—present in some *C. burnetii* strains causing acute Q fever in humans [21]—was present in this genotype II+ strain. Genotype II+ has been previously reported in ticks (Slovak Republic), sheep (Germany, Spain) and humans (Italy) and it appears to be the most widely distributed (RLB) genotype in Europe [19]; nevertheless, the absence of molecular epidemiology studies in wildlife and in particular in micromammals makes the understanding of potential cross-species transmission difficult. Typing pathogens circulating in healthy wildlife could be partly constrained by pathogen burden in tissues [18,19]. This may occur more frequently in enzootic pathogens that have a large history of co-evolution with their hosts and replicate at a lower ratio in hosts than epidemic pathogens [22]. This, however, does not detract these hosts from playing relevant roles in the life cycle of pathogens. Currently, no clinical consequence of *C. burnetii* infection has been reported or noticed in infected micromammals; none of the micromammals surveyed in this study had symptoms compatible with Coxiellosis. A significant number of the infected individuals did indeed display very low levels of *C. burnetii* DNA in the spleen (69 of the 79 qPCR-positives had Ct values close to the negative threshold), perhaps indicative of past or subclinical infections. We did indeed obtain partial RLB typing results for a major part of the analysed samples, but only one could be completely typed. 

In recent decades, zoonotic emerging infectious diseases have advanced positions to become one of the most worrisome threats to human, livestock and wildlife health [23,24]; SARS-CoV-2 has an animal origin, and it has been able to cross the inter-species barrier to emerge as the most devastating human pandemic of our time [25]. This may be owed to changes in the patterns of interaction between domestic animals, wildlife and humans [4] that are most probably occurring due to human influences on habitats, biodiversity and the climate. These changing patterns may also be behind the re-emergence of enzootic zoonoses such as Q fever that, although with a lower pandemic potential, may become a serious health problem. Our study contributes to unravelling the potential future threats of the re-emergence of *C. burnetii* infections of wildlife origin by informing about the potential implication of micromammals in the interspecific exchange of the pathogen. It also highlights the relevance of opportunistic sample collection in providing basic descriptive information useful to design future epidemiological studies. Findings reveal the occurrence of infections by *C. burnetii* in different species of wild micromammals in Spain as well as the presence of genotypes shared with humans, ticks and domestic animals and reported in different European countries [18,19].

In previous European studies, *C. burnetti* DNA was found in spleen samples of *R. rattus*, *R. norvegicus*, *Mus musculus*, *A. flavicollis* and *A. sylvaticus* from Cyprus, Germany, The Netherlands, Italy, Slovakia and Spain (Table 3). Prevalence ratio in medium-to-large sized studies ranged 0.6–23.5%. Other micromammal species such as the bank vole, the common vole and the common shrew that were object of medium-to-large surveys did not show infection with *C. burnetii*.

European studies on *C. burnetii* have mainly focused on the most widespread and abundant micromammal species in Europe such as *A. sylvaticus*, *Myodes glareolus*, *A. flavicollis*, *M. arvalis* and *Rattus* sp. The genus *Rattus* has been found consistently positive to *C. burnetii* DNA in all the studies performed in Europe, so rats are currently considered as true *C. burnetii* reservoirs [6,8,9]. *Coxiella burnetii* DNA was also detected in *R. rattus* in southern Spain in this study, so rats can be relevant hosts for *C. burnetii* in the Iberian Peninsula as well. These may play also a relevant role in the exchange of *C. burnetii* at the wildlife–livestock–human interface because rats live in natural, peridomestic and urban environments. In most of the studies in which *C. burnetii* DNA has been detected in micromammals, these were captured in peridomestic areas where domestic ruminants (known *C. burnetii* reservoirs) were present. The micromammals surveyed for this study came from wild environments where the direct interaction with domestic ruminants is low or inexistent. Wild ungulates and other proven wild reservoirs of *C. burnetii* [13,14] may be frequent in some of these areas but have been found to host mainly strains of the genotypes I and VII [19]. Our results cannot confirm that all the qPCR positive micromammal species found in this study are reservoirs for *C. burnetii*, but qPCR positive uterus samples from common voles were previously found [19] and a positive vaginal swab from *M. arvalis* was also found in this study, therefore demonstrating that *M. arvalis* is able to replicate and shed *C. burnetii*. An additional observation supporting the potential reservoir role of *M. arvalis* was the diagnosis of acute Q fever in one of the researchers (and author in this study) that participated in the survey of common voles. This person presented to a local medical practitioner in April 2012 with high fever and malaise a week after finishing a vole survey in northwestern Spain, and it was presumably diagnosed with flu. Some days later, the patient had persistent high fever and visited a hospital in northern Spain where Q fever was confirmed by detecting high titres (1/320) of specific *C. burnetii* IgG antibodies in an indirect immunofluorescent assay (IFA) and by a positive result in a specific IgM ELISA test for *C. burnetii*. The patient was negative to brucellosis, Lyme disease, tularaemia, bartonellosis and hepatitis B in a differential diagnosis approach of the probable infectious causes of persistent fever in a patient exposed to wildlife. This person had no history of exposure to livestock and other wildlife environments and had spent several weeks surveying voles before the onset of the symptoms. Whether this case was related to vole handling or was the consequence of exposure to contaminated aerosols from livestock or other wildlife shedding *C. burnetii* could not be determined.

The potential implication of Spanish micromammal species in the ecology of *C. burnetii* is based on some of the findings (in this and in other studies) such as the detection of specific *C. burnetii* antibodies in a wide diversity of micromammal species, including common vole, rats, house mouse, wood mouse and yellow-necked field mouse [4] that demonstrate susceptibility to pathogen infection. Further support comes from finding *C. burnetii* DNA in spleen samples of some micromammal species that indicates that a bacteraemia following replication did occur. In addition, *C. burnetii* DNA was found in the reproductive tract and vaginal secretions of *M. arvalis*, further supporting a potential efficient role in *C. burnetii* replication and transmission. The detection of *C. burnetii* DNA in rat faeces in other studies [36] also points out that (at least) rats also allow replication of *C. burnetii* and shedding. 

The common vole experiences cyclic population density peaks under highly favourable environmental conditions that do indeed drive the exchange of zoonotic multi-host pathogens at the wildlife–human interface, e.g., *Francisella tularensis* [16]. However, in contrast to the spill-over role that *M. arvalis* plays in the transmission of *F. tularensis*, it may play a true reservoir role for *C. burnetii* and maintain it independently from the co-occurrence of other relevant hosts, e.g., lagomorphs or ungulates. Tularaemia is an emerging disease that currently is present only in the northern half of mainland Iberian Peninsula where it affects humans, wild lagomorphs and micromammals [37]. In this study, we observed in common voles from this area a slight increase in *C. burnetii* infection prevalence (unpublished data) from 2012 (low vole abundance) to 2014 (vole density peak in the study area; [38]), demonstrating persisting pathogen prevalence under conditions of contrasted host density (up to 100-fold), and supporting its role as true reservoir in clear contrast with tularaemia that practically disappears from vole populations in low abundance years [16]. During vole population outbreaks, infection rates and environmental contamination with *C. burnetii* could peak, given extremely high abundance of voles around or even inside villages, which could increase Q fever and Coxiellosis risks for humans and animals, respectively. This may be perhaps partly reflected in the observed higher incidence of Coxiellosis in sheep farms [39] during years of high vole density (2010–2011) in the region where *M. arvalis* were surveyed for this study [38]. *C. burnetii* transmission to livestock could be relevant in years or areas with high density of voles, but more precise information and longer time series should be required to confirm this possibility and any potential link among voles and domestic ruminants in *C. burnetii* exchange. The human infection case reported above indicates that frequent and tight contact with *M. arvalis* is a risk factor potentially promoting Q fever, so recommendations about avoiding contact with these animals disseminated to rural populations in vole outbreak years are highly advisable. The participation of potential coexisting micromammal reservoir species, e.g., *A. sylvaticus* and *C. russula*, would add complexity to understanding the factors shaping *C. burnetii* transmission risks in a sylvatic cycle [16]. Thus, sanitary recommendations to avoid contact with small mammals should be extended in general terms in the Iberian Peninsula. 

## 4. Conclusions

We can conclude that this first approach provides evidence supporting the fact that there are several micromammal species that can be potential reservoirs of *C. burnetii*. Abundant and widespread species in the Iberian Peninsula, e.g., rats, wood mouse and white-toothed shrew, as well as species experiencing drastic cyclic demographic outbreaks, i.e., the common vole, might be relevant in the maintenance of wild-type *C. burnetii* strains that can be a matter of concern for animal and human health authorities. 

## Figures and Tables

**Figure 1 animals-11-00654-f001:**
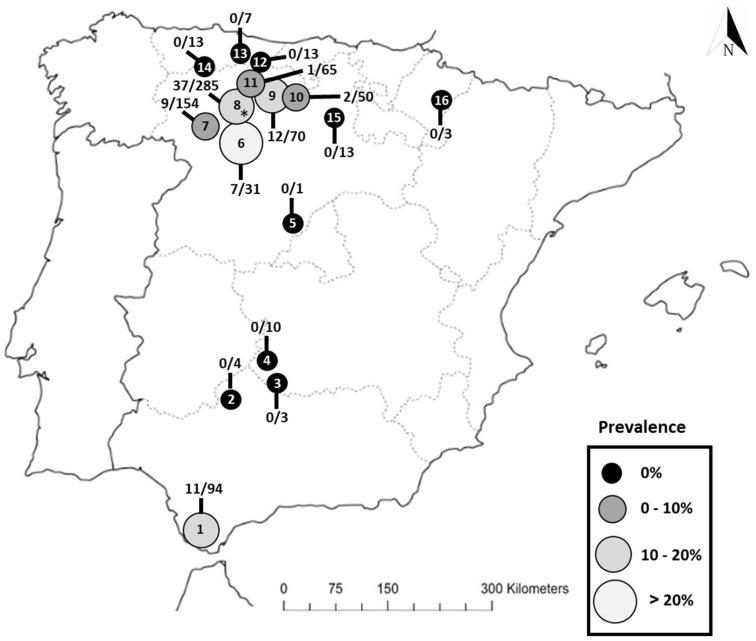
Spatial distribution and prevalence of *Coxiella burnetii* DNA in spleen samples from micromammals. Each dot (overall sample size included) represents a surveyed population of micromammals. The numbers shown per location indicate the number of positive samples with respect to local sample size (positives/total). The size and color of the dots show population prevalence of *C. burnetii* infection as detailed in the legend. The asterisk (*) in a dot indicates that a *C. burnetii* genotype was obtained in this population.

**Table 1 animals-11-00654-t001:** Species of micromammals surveyed per study location as shown in Figure 1: *Apodemus flavicollis* (Af), *Apodemus sylvaticus* (Ap), *Arvicola terrestris* (At), *Crocidura russula* (Cr), *Eliomys quercinus* (Eq), *Microtus arvalis* (Ma), *Mus musculus* (Mm), *Mus spretus* (Ms), *Rattus rattus* (Rr), *Sciurus vulgaris* (Sv). The PCR-positive (p) vs. the total number (n) of samples per micromammal species (p/n) and location is shown. In addition, the year(s) and month(s) of sampling, the habitat type, the presence of other co-existing animal species as well as the existence of evidence of previous detection of C. burnetii in non-micromammal species in the location are included. n.a. = data not available.

Location Reference	Micromammal Species Surveyed	Sampling Period	Habitat Type	Co-Habitation with Other Animals	Previous DNA Detection of *C. burnetii* in Other Species
**1**	As (7/33), Cr (1/8), Eq (0/2), Ms (0/4), Rr (3/47)	2013 (January, June, July, December)	Natural Mediterranean scrubland with large areas of irrigated prairies.	Wildlife	Yes [13,14]
**2**	As (0/1), Cr (0/2), Ms (0/1)	2003 (April)	Natural Mediterranean scrubland with Savannah-like areas	Wildlife	Yes [13,18]
**3**	As (0/1), Mm (0/2)	2003 (June)	Natural Mediterranean scrubland	Wildlife	Yes [18]
**4**	As (0/5), Ms (0/5)	2004 (April)	Natural Mediterranean scrubland	Wildlife	n.a.
**5**	Sv (0/1)	2008 (December)	Natural Mediterranean scrub with pinelands	Wildlife	Yes [19]
**6**	As (0/1), Ma (7/30)	2013 (April) 2014 (May)	Agricultural areas	Occasionally sheep	n.a.
**7**	As (0/18), Ma (9/134), Ms (0/2)	2012 (March–July, October)	Agricultural areas	Occasionally sheep	n.a.
**8**	As (4/45), Cr (0/5), Ma (33/232), Ms (0/3)	2012 (January–November) 2013 (March–May) 2014 (May)	Agricultural areas	Occasionally sheep	Yes [19]
**9**	As (1/13), Cr (0/2), Ma (11/51), Ms (0/4)	2011 (November–December) 2012 (January–April)	Agricultural areas	Occasionally sheep	n.a.
**10**	As (0/4), Cr (0/1), Ma (2/44), Ms (0/1)	2012 (September)	Agricultural areas	Occasionally sheep	n.a.
**11**	As (0/2), At (0/1), Cr (1/3), Ma (0/59)	2012 (August, September, November)	Agricultural areas	Occasionally sheep	n.a.
**12**	As (0/1), Ma (0/12)	2012 (October)	Agricultural areas	Wildlife and extensive cattle breeding	n.a.
**13**	Af (0/2), As (0/4), Mm (0/1),	2003 (July) 2013 (July)	Atlantic forest interspersed with scrublands and prairies	Wildlife and extensive cattle breeding	Yes [19]
**14**	Cr (0/3), Ma (0/10)	2012 (August)	Atlantic forest interspersed with scrublands and prairies	Wildlife and extensive cattle breeding	n.a.
**15**	As (0/10), Cr (0/1), Mm (0/2),	2003 (June)	Atlantic forest interspersed with scrublands and prairies	Wildlife	n.a.
**16**	Cr (0/1), Ms (0/2)	2012 (December)	Steppe and Mediterranean vegetation (“Bardenas Reales”)	Wildlife	Yes [13]

**Table 2 animals-11-00654-t002:** qPCR results by micromammal species object of survey in this study for each location. Species are displayed along with sample size (*n*) and the number of qPCR positive samples (PCR positives). In addition, 95% exact confidence intervals are shown within brackets.

Location Reference	Species	*n*	PCR Positives	% PCR Positive
1	*Apodemus sylvaticus*	33	7	21.2 (10.7–37.8)
	*Crocidura russula*	8	1	12.5 (2.2–47.1)
	*Eliomys quercinus*	2	0	0.0 (0.0–65.8)
	*Mus spretus*	4	0	0.0 (0.0–48.9)
	*Rattus rattus*	47	3	6.38 (2.2–17.2)
2	*Apodemus sylvaticus*	1	0	0.0 (0.0–79.3)
	*Crocidura russula*	2	0	0.0 (0.0–65.8)
	*Mus spretus*	1	0	0.0 (0.0–79.3)
3	*Apodemus sylvaticus*	1	0	0.0 (0.0–79.3)
	*Mus musculus*	2	0	0.0 (0.0–65.8)
4	*Apodemus sylvaticus*	5	0	0.0 (0.0–43.4)
	*Mus spretus*	5	0	0.0 (0.0–43.4)
5	*Sciurus vulgaris*	1	0	0.0 (0.0–79.3)
6	*Apodemus sylvaticus*	1	0	0.0 (0.0–79.3)
	*Microtus arvalis*	30	7	23.3 (11.8–40.9)
7	*Apodemus sylvaticus*	18	0	0.0 (0.0–17.6)
	*Microtus arvalis*	134	9	6.7 (3.6–12.3)
	*Mus spretus*	2	0	0.0 (0.0–65.8)
8	*Apodemus sylvaticus*	45	4	8.9 (3.5–20.7)
	*Crocidura russula*	5	0	0.0 (0.0–43.4)
	*Microtus arvalis*	232	33	14.2 (10.3–19.3)
	*Mus spretus*	3	0	0.0 (0.0–56.1)
9	*Apodemus sylvaticus*	13	1	7.7 (13.7–33.3)
	*Crocidura russula*	2	0	0.0 (0.0–65.8)
	*Microtus arvalis*	51	11	21.6 (12.5–34.6)
	*Mus spretus*	4	0	0.0 (0.0–48.9)
10	*Apodemus sylvaticus*	4	0	0.0 (0.0–48.9)
	*Crocidura russula*	1	0	0.0 (0.0–79.3)
	*Microtus arvalis*	44	2	4.5 (1.3–15.1)
	*Mus spretus*	1	0	0.0 (0.0–79.3)
11	*Apodemus sylvaticus*	2	0	0.0 (0.0–65.8)
	*Arvicola terrestris*	1	0	0.0 (0.0–79.3)
	*Crocidura russula*	3	1	33.3 (6.1–79.2)
	*Microtus arvalis*	59	0	0.0 (0.0–6.1)
12	*Apodemus sylvaticus*	1	0	0.0 (0.0–79.3)
	*Microtus arvalis*	12	0	0.0 (0.0–24.2)
13	*Apodemus flavicollis*	2	0	0.0 (0.0–65.8)
	*Apodemus sylvaticus*	4	0	0.0 (0.0–48.9)
	*Mus musculus*	1	0	0.0 (0.0–79.3)
14	*Crocidura russula*	3	0	0.0 (0.0–56.1)
	*Microtus arvalis*	10	0	0.0 (0.0–27.7)
15	*Apodemus sylvaticus*	10	0	0.0 (0.0–27.7)
	*Crocidura russula*	1	0	0.0 (0.0–79.3)
	*Mus musculus*	2	0	0.0 (0.0–65.8)
16	*Crocidura russula*	1	0	0.0 (0.0–79.3)
	*Mus spretus*	2	0	0.0 (0.0–65.8)

**Table 3 animals-11-00654-t003:** Review of current evidences of *Coxiella burnetii* DNA detection in European micromammal species.

Common Name	Scientific Name	Country	Pos/N (Prev)	Reference
Bank vole	*Myodes glareolus*	Austria	0/40 (0.0)	[26]
		Croatia	0/43 (0.0)	[27]
		Italy	0/42 (0.0)	[28]
		Slovakia	0/23 (0.0)	[29]
			0/239 (0.0)	[30]
		Spain	0/6 (0.0)	[31]
		Czech Republic/Germany	0/78 (0.0)	[32]
Black rat	*Rattus rattus*	Netherlands	5/166 (3.0)	[6]
		Spain	3/47 (6.4)	This study
Brown rat	*Rattus norvegicus*	Germany	7/524 (1.3)	[8]
		Netherl.	8/164 (4.8)	[6]
Brown/black rat	*Rattus* spp.	Cyprus	32/136 (23.5)	[33]
		Spain	3/3 (100.0)	[20]
Common vole	*Microtus arvalis*	Austria	0/15 (0.0)	[26]
		Croatia	0/4 (0.0)	[27]
		Germany	0/109 (0.0)	[34]
		Slovakia	0/3 (0.0)	[29]
			0/19 (0.0)	[30]
		Czech Republic/Germany	0/148 (0.0)	[32]
		Spain	62/572 (10.9)	This study
Eurasian Harvest Mouse	*Micromys minutus*	Slovakia	0/1 (0.0)	[30]
European Pine Vole	*Microtus subterraneus*	Slovakia	0/1 (0.0)	[30]
European Water Vole	*Arvicola terrestris*	Spain	0/1 (0.0)	This study
		Germany	0/3 (0.0)	[34]
Field vole	*Microtus agrestis*	Croatia	0/1 (0.0)	[27]
		Czech Republic/Germany	0/1 (0.0)	[32]
Hazel dormouse	*Muscardinus avellanarius*	Croatia	0/1 (0.0)	[27]
House mouse	*Mus musculus*	Spain	2/28 (7.1)	[31]
		Spain	8/61 (13.1)	[35]
		Spain	0/10 (0.0)	This study
Long-tailed field mouse	*Apodemus sylvaticus*	Austria	0/26 (0.0)	[26]
		Croatia	0/3 (0.0)	[27]
		Italy	2/101 (19.8)	[28]
		Slovakia	0/3 (0.0)	[29]
			0/3 (0.0)	[30]
		Czech Republic/Germany	0/6 (0.0)	[32]
		Germany	0/2 (0.0)	[34]
		Spain	1/162 (0.6)	[31]
		Spain	12/138 (8.7)	This study
Striped Field Mouse	*Apodemus agrarius*	Croatia	0/54 (0.0)	[27]
		Czech Republic/Germany	0/2 (0.0)	[32]
Yellow-necked field mouse	*Apodemus flavicollis*	Austria	0/29 (0.0)	[26]
		Croatia	0/131 (0.0)	[27]
		Slovakia	1/38 (2.6)	[29]
			0/401 (0.0)	[30]
		Czech Republic/Germany	0/48 (0.0)	[32]
		Germany	0/3 (0.0)	[34]
		Spain	0/3 (0.0)	[31]
		Spain	0/2 (0.0)	This study
Common Shrew	*Sorex araneus*	Czech Republic/Germany	0/30 (0.0)	[32]
Crowned Shrew	*Sorex coronatus*	Czech Republic/Germany	0/7 (0.0)	[32]
Eurasian Pygmy Shrew	*Sorex minutus*	Czech Republic/Germany	0/1 (0.0)	[32]
White-toothed Shrew	*Crocidura russula*	Spain	2/26 (7.7)	This study

## Data Availability

The data presented in this study are available in this article and Appendix A.

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
