# Peer review of "Investigating the Role of Micromammals in the Ecology of Coxiella burnetii in Spain"

_animals, 2021, doi:10.3390/ani11030654_

Round 1

Reviewer 1 Report

Dear authors,

thanks for the revised manuscript. Some very few language issues are still present, but other than that all very well adapted.

Best regards

Author Response

We are grateful to the Reviewer 1 for the time and effort in the evaluation of our study.

Reviewer 2 Report

I congratulate González-Barrio et al. on a clear and well written paper. The recent changes have further improved the manuscript and fully addressed the minor changes highlighted in the previous review. I have no further comments or suggested changes.

Author Response

We are grateful to the Reviewere 2 for the time and effort in the evaluation of our study.

This manuscript is a resubmission of an earlier submission. The following is a list of the peer review reports and author responses from that submission.

Round 1

Reviewer 1 Report

This manuscript presents the results of real-time PCR analyses performed to detect DNA of C. burnetii in the spleen (n=814) and vaginal swabs (n=130) from diverse micromammals species sampled in diverse localities of Spain between 2004 and 2014.

Main comments

Such descriptive studies are important to better understand the circulation of C. burnetii within wild species and micromammals in particular. However, to be really informative, they need to be placed into context, particularly regarding the presence and infection status of other host species, both wild and domestic. Indeed, in the absence of extensive data about the whole ecological system, it is not possible to really provide “new epidemiological insight on Coxiella burnetii ecology” as suggested in the manuscript title.

The present study gathers data sampled from diverse sampling sites, at various periods of time, without providing any detail about the contexts of these respective studies. It is also unclear whether parts of these results have previously been published by the authors (e.g. Jado et al. 2012, Gonzales-Barrio et al. 2016).

Overall, demonstrating that a species, or a group of species, is a reservoir of a multi-host disease such as Q fever is highly complex. The discussion about demonstrating that an animal species is a reservoir, solely based on the reference “Wobeser 1994”, needs to be updated with more recent references such as Haydon et al. EID 2002 (Identifying Reservoirs of Infection: A Conceptual and Practical Challenge) and Viana et al. TREE 2014 (Assembling evidence for identifying reservoirs of infection). In the absence of more detailed data, the current discussion about the “potential role of reservoir” relies on limited and mainly speculative data.

Detailed remarks

Simple summary

The sentence “According to these results, micromammals seem also relevant wild hosts for C. burnetii and potential risk factors to consider when tracing the origin of human Q fever and animal coxiellosis cases in Europe” needs rewording: the results of the study are consistent with previous data showing that micromammals can be infected by C. burnetii

Abstract

The sentence “According to these results, micromammals seem also relevant wild hosts for C. burnetii” needs rewording because it was already known that micromammals are hosts for C. burnetii

Intro

Line 66: C. burnetii infection has been neglected in wildlife: more explanations are needed to clarify what the authors mean: there are many publications about C. burnetii in wildlife (including publications much older than those cited by the authors). In this context, what are really the motivations to consider that Q fever has been “neglected” in wildlife?

Line 82-85 : “it would however be expected that the wide diversity of micromammals in the country as well as the current expansion of some ‘pest’ rodent species would have an impact in the sylvatic cycle of the bacterium”: I do not see why the authors make the hypothesis that the “diversity” of these species would impact the sylvatic cycle? Please develop/explain or remove

Material and methods

The different contexts in which the micromammals were sampled need to be specified (type of environment, domestic and wild species previously described as Q fever reservoir present or not, Q fever data available for other species or not…). Why were these sampling sites chosen?

The time of year would also be relevant to mention, particularly in reference to the results obtained from vaginal samples (parturient females?) and to the birthing season of other species that may be present in the considered ecosystems (ewes in particular).

Line 109-110: some individuals were euthanized: what were the inclusion criteria? Why mentionning the other animals (released??)? Please explain

Line 112: were precautions undertaken to avoid and/or monitor potential contact/aerial cross-contaminations during the necropsies?

Line 113: which swabs were used? Were they collected from live or dead females? Was the lactating or gestating status checked?

Line 116: found dead instead of death

Lines 103-117: were any precautions taken to secure the manipulations and prevent contaminations of the persons handling the animals?

Lines 121-130: the protocol used to extract DNA from the spleen samples is not sufficiently detailed

Line 129: how cross-contamination was checked during DNA extraction? Were negative control tubes kept open or closed during the extraction process? How many were used?

Line 131: spleen and swab samples (?)

Lines 134-136: are cluster analyses (and related phylogenetic inferences) based on RLB really robust, particularly since the genome of C. burnetii contains many insertion sequences? Why not considering other typing methods based on sequencing for instance? Something to discuss perhaps?

Results and discussion

Lines 149-152: is there really a rationale for using confidence intervals given the fact that the animals sampled do not (always) belong to a same population?

Lines 148-153: Although the sex and age of the animals have been considered according to the material and method section, the corresponding results are not presented.

Lines 155-156: the relationship between “acute” Q fever and the “ada gene” is debated, see Frangoulidis et al. Plos One 2013 Microevolution of the Chromosomal Region of Acute Disease Antigen A (adaA) in the Query (Q) Fever Agent Coxiella burnetii

Lines 160-161: “the limitation of the bacterial burden in host samples caused by the evolution of the infection as previously described”: unclear, please explain

Line 163: “A major part of the infected individuals displayed very low levels of C. burnetii DNA (…) perhaps indicative of past or subclinical infection”: these low levels are not in favor of the fact that these species represent an important reservoir of the disease and should be taken into consideration in the discussion

Lines 173-174: “Our study contributes to unravelling the potential future threats of re-emergence of C. burnetii infections of wildlife origin “: excessive conclusion, this sentence should be removed

Lines 174-175: “and it is a unique national-scale molecular survey of the pathogen in wild micromammals”: because the sampling scheme is opportunistic, on large time frame (10 years) and not representative of any micromammals population, the term “unique national scale molecular survey” seems excessive

Lines 177-178: “Major findings reveal (…) the presence of genotypes shared with humans, ticks and domestic animals and reported in different European countries [19,22])”: to remove because only one sample from this study was successfully genotyped

Table 2: I spotted a mistake regarding the results of Microtus arvalis, when referring to the study from Pluta et al. (Germany, 0/119): there were actually 109 M. arvalis tested (other species were M. Arvicola terrestris (n = 3), Apodemus flavicollis (n = 3), and Apodemus sylvaticus (n = 2).

Lines 196-197: “The wild micromammals surveyed for this study came mostly from areas where livestock is absent or, if present, only present at specific times in the year”: this point clearly requires further details

Lines 203-204: “An additional observation supporting the potential reservoir role of M. arvalis was the diagnosis of acute Q fever in one of the researchers”: further details are needed to exclude the possibility of contamination from other species that may have excreted C. burnetii in the environment where the field campaigns took place. Also, details about the conditions of manipulations (use of FFP2 or FFP3 masks, gloves, hood…) are needed? Any event of cutting during the dissections?

Lines 214 and beyond: as specified in the major comment section, the discussion about “what it takes to be a reservoir” needs to be more developed and the conclusions of the manuscript adapted accordingly

Author Response

Reviewers’ comments

Reviewer 1

This manuscript presents the results of real-time PCR analyses performed to detect DNA of C. burnetii in the spleen (n=814) and vaginal swabs (n=130) from diverse micromammals species sampled in diverse localities of Spain between 2004 and 2014.

Main comments

Such descriptive studies are important to better understand the circulation of C. burnetii within wild species and micromammals in particular. However, to be really informative, they need to be placed into context, particularly regarding the presence and infection status of other host species, both wild and domestic. Indeed, in the absence of extensive data about the whole ecological system, it is not possible to really provide “new epidemiological insight on Coxiella burnetiid ecology” as suggested in the manuscript title.

Authors’ Response (AR): We highly appreciate the effort of the reviewer in thoroughly checking the quality of the study and the manuscript. We are very grateful for the relevant points he/she addresses for discussion and the recommendations. We’ve carefully checked any comment and addressed them with changes in the manuscript when considered relevant for the study or provided a comprehensive sum of reasons when recommendations are not followed.

‘Informative’ means that it provides information and ‘insights’ are novel findings in a particular topic by definition. Our study (despite the opportunistic survey nature and the limitations caused by unbalances in sample size) provides the reader with information on novel findings over a large area with no information available in the EU country with the highest number of Q fever notified cases. We have slightly modified the title, so now it reads ‘Investigating the role of micromammals in the ecology of Coxiella burnetii in Spain’.

The present study gathers data sampled from diverse sampling sites, at various periods of time, without providing any detail about the contexts of these respective studies. It is also unclear whether parts of these results have previously been published by the authors (e.g. Jado et al. 2012, Gonzales-Barrio et al. 2016).

AR: The reviewer is partly right. He/she is right in pointing out that this study is the result of opportunistically gathering samples of wild micromammals from different studies; we have tried to better highlight the opportunistic nature of the survey in the manuscript. We however cannot find the relevance of showing the readers the specific context (specific goals behind any study) in which these samples were collected and how this is informing them to better interpret the descriptive findings. We agree in that it is relevant showing the readers that the samples were collected at different time periods and the length of the periods in which these samples were collected, so they can estimate whether a single epidemiological population of a particular species of micromammals was surveyed in a single time point or along a long period of time. The timing of sample collection in each of the study populations could obviously have effects on infection prevalence parameters as enzootic infections are dynamic in wildlife (see González-Barrio et al. (2015) Frontiers in Veterinary Science). A new table displaying the timing of sample collection for any of the epidemiological populations shown in Figure 1, a rough description of the habitat of the study area, the presence of other potential known C. burnetii hosts and if C. burnetii has been previously detected in other hosts has been included in the revised version of the manuscript.

Jado et al. (2012) BMC Microbiology does not present any of the findings in this study. González-Barrio et al. (2016) Environmental Microbiology Reports, included some (n=9) PCR positive samples from micromammals for genotyping with negative results; these 9 positive animals are also included in this study.

Overall, demonstrating that a species, or a group of species, is a reservoir of a multi-host disease such as Q fever is highly complex. The discussion about demonstrating that an animal species is a reservoir, solely based on the reference “Wobeser 1994”, needs to be updated with more recent references such as Haydon et al. EID 2002 (Identifying Reservoirs of Infection: A Conceptual and Practical Challenge) and Viana et al. TREE 2014 (Assembling evidence for identifying reservoirs of infection). In the absence of more detailed data, the current discussion about the “potential role of reservoir” relies on limited and mainly speculative data.

AR: We agree with the reviewer and we are aware of the relevance of community ecology and host population assemblages in maintaining pathogens (not diseases) as described by Haydon et al. and Viana et al. We however also agree in the basic postulates described by Gary Wobeser to consider a species as a reservoir in essence since this is also relevant for pathogen management. Both perceptions are compatible; the first better describes the ecological population and community processes leading to multi-host pathogen maintenance in specific habitats/areas/populations/communities whereas Wobeser postulates apply to the expected potential role of a particular host species. We herein adopt the Wobeser postulates as our focus was not the community of hosts at particular populations and specific scenarios of host community assemblages, but rather particular species as a whole. We however have slightly modified this section as the discussion on this particular issue can be built with the observations of the study.

Detailed remarks

Simple summary

The sentence “According to these results, micromammals seem also relevant wild hosts for C. burnetii and potential risk factors to consider when tracing the origin of human Q fever and animal coxiellosis cases in Europe” needs rewording: the results of the study are consistent with previous data showing that micromammals can be infected by C. burnetiid

AR: We are aware of previous findings and have summarized them in a recent review (see González-Barrrio et al. 2019 Transboundary and Emerging Diseases). We intended not to show that our findings are the first claim to micromammals being relevant in C. burnetii ecology, so we have accordingly rewritten this sentence.

Abstract

The sentence “According to these results, micromammals seem also relevant wild hosts for C. burnetii” needs rewording because it was already known that micromammals are hosts for C. burnetii

AR: See comment above.

Intro

Line 66: C. burnetii infection has been neglected in wildlife: more explanations are needed to clarify what the authors mean: there are many publications about C. burnetii in wildlife (including publications much older than those cited by the authors). In this context, what are really the motivations to consider that Q fever has been “neglected” in wildlife?

AR: The term ‘neglected’ applies to something that is not given proper or necessary care or attention. The reviewer would agree in that attention to the relationship wildlife-C. burnetii has been neglected in the scientific literature. Data support this perception as there are far fewer publications (about 200 in SCOPUS) on this subject compared to livestock (nearly 2,000 publications). We have clearly stated what led us considering this as a neglected topic in wildlife health research.

Line 82-85 : “it would however be expected that the wide diversity of micromammals in the country as well as the current expansion of some ‘pest’ rodent species would have an impact in the sylvatic cycle of the bacterium”: I do not see why the authors make the hypothesis that the “diversity” of these species would impact the sylvatic cycle? Please develop/explain or remove

AR: This perception is based on the diversity of micromammal species apparently (from literature records) contributing to C. burnetii maintenance. Therefore, as higher the diversity of species and overall numbers, higher the chances for C. burnetii to find hosts and replicate. Further, Microtus arvalis is considered a pest species whose demographic outbreaks drive pathogen prevalence and exchange, so it makes sense to employ the reasoning method of analogy to hypothesize that their expansion would also impact C. burnetii ecology.  We have highlighted the basis for this hypothesis in the text.

Material and methods

The different contexts in which the micromammals were sampled need to be specified (type of environment, domestic and wild species previously described as Q fever reservoir present or not, Q fever data available for other species or not…). Why were these sampling sites chosen?

AR: This information has been gathered in a new table. No specific criteria were applied for inclusion of micromammal samples in this study. We included all the samples that could be gathered to estimate C. burnetii infection at the study scale.

The time of year would also be relevant to mention, particularly in reference to the results obtained from vaginal samples (parturient females?) and to the birthing season of other species that may be present in the considered ecosystems (ewes in particular).

AR: This information has also been included in Table 1. In contrast to ungulates that naturally concentrate births in a season within a year, micromammals breed along periods of favourable environmental conditions with no clear birth seasonal clusters. Genital swabs were collected at different times along the year in live captured M. arvalis but the reproductive status could not be recorded visually.

Line 109-110: some individuals were euthanized: what were the inclusion criteria? Why mentionning the other animals (released??)? Please explain

AR: No specific inclusion criteria were followed to select individuals from those captured in traps to be euthanized, it was random. We state that only ‘some’ individuals were euthanized to show that we did not euthanize every captured animal.

Line 112: were precautions undertaken to avoid and/or monitor potential contact/aerial cross-contaminations during the necropsies?

AR: Yes, the small size of the animals allowed performing the necropsies in BSL2 cabinets to avoid cross-contaminations. We employed disposable material (e.g., gloves), sterile containers for samples and sterilized every single tool employed in the necropsy between different samples and individuals chemically (disinfectants) and/or physically (gas flame).

Line 113: which swabs were used? Were they collected from live or dead females? Was the lactating or gestating status checked?

AR: We employed metal viscose swabs with AMIES medium (added to the text). The swabs were collected from live captured M. arvalis females and the reproductive status was not recorded.

Line 116: found dead instead of death

AR: Thank you; it has been corrected.

Lines 103-117: were any precautions taken to secure the manipulations and prevent contaminations of the persons handling the animals?

AR: Swabs were collected from M. arvalis in 2012, when Q fever was not of mandatory notification in humans in Spain, so precautions only consisted in the manipulation of animals with protection gloves. No mask and no glasses were employed during the surveys.

Lines 121-130: the protocol used to extract DNA from the spleen samples is not sufficiently detailed

AR: Ok, we have further detailed the protocol in the text of the revised version.

Line 129: how cross-contamination was checked during DNA extraction? Were negative control tubes kept open or closed during the extraction process? How many were used?

AR: As stated in the text, sample cross-contamination was discarded by including a negative control of extraction (Nuclease free water) every ten samples. The controls were treated as the rest of the samples during purification. We have been running this protocol in our labs for 15 years for several different samples, studies and nucleic acid purification systems with very rare focal cross-contaminations events.

Line 131: spleen and swab samples (?)

AR: Corrected, thank you.

Lines 134-136: are cluster analyses (and related phylogenetic inferences) based on RLB really robust, particularly since the genome of C. burnetii contains many insertion sequences? Why not considering other typing methods based on sequencing for instance? Something to discuss perhaps?

AR: We have carried out RLB typing analyses previously that resulted in the successful identification of host clusters for particular types, so we chose this typing method to be able to compare with previous results. PCR-RLB genotypes described by González-Barrio et al., 2016 (doi: 10.1111/1758-2229.12431) in wildlife provided the same host clustering findings obtained with MLVA typing techniques (doi: 10.1007/s00248-016-0786-9). We are currently moving towards full genome sequencing as a more robust tool to infer basic ecological traits of different C. burnetii strains.

Results and discussion

Lines 149-152: is there really a rationale for using confidence intervals given the fact that the animals sampled do not (always) belong to a same population?

AR: Confidence intervals estimate the precision of a prevalence. We find relevant showing the precision of our estimates at population levels spatially and metapopulation levels for host species.

Lines 148-153: Although the sex and age of the animals have been considered according to the material and method section, the corresponding results are not presented.

AR: We indeed registered further information from every individual during necropsies and preserved several other samples. We however did not carry over any analysis on the influence of specific population and individual risk factors over infection risk or prevalence due to unbalances. We have deleted the sentence to avoid confusing the readers on what expecting to find in the results.

Lines 155-156: the relationship between “acute” Q fever and the “ada gene” is debated, see Frangoulidis et al. Plos One 2013 Microevolution of the Chromosomal Region of Acute Disease Antigen A (adaA) in the Query (Q) Fever Agent Coxiella burnetii

AR: Yes, the reviewer is correct; this has been modified in the text.

Lines 160-161: “the limitation of the bacterial burden in host samples caused by the evolution of the infection as previously described”: unclear, please explain

AR: We have tried to clarify our meaning by modifying this sentence.

Line 163: “A major part of the infected individuals displayed very low levels of C. burnetii DNA (…) perhaps indicative of past or subclinical infection”: these low levels are not in favor of the fact that these species represent an important reservoir of the disease and should be taken into consideration in the discussion

AR: Why? Would surveys performed on sheep several months after breeding find high levels of C. burnetii being shed in feces, milk or vaginal secretions? Would this mean that sheep are not efficient reservoirs of C. burnetii? Furthermore, why a reservoir needs to be shedding large burdens of the bacteria to be an effective reservoir when a single phase I bacterium is able to run an effective infection in the host? Would a 50-250g host be able to shed the large burdens of Coxiella that a 40 to 500 kg ungulate is able to shed? The rate of effective interactions leading to pathogen exchange between infected and susceptible individuals is what in the end determines the epidemiologic relevance of a species (community or group) with respect the maintenance of a pathogen.

Lines 173-174: “Our study contributes to unravelling the potential future threats of re-emergence of C. burnetii infections of wildlife origin “: excessive conclusion, this sentence should be removed

AR: We do not agree with the perception of the reviewer because our study is contributing with unknown information on the status of C. burnetii infection in different micromammal species in Spain, a country with high incidence of Q fever and Coxiellosis. Our study is also providing relevant information on the status of C. burnetii in M. arvalis in north-central Spain where the species experiments sudden demographic changes that are proven to drive the risk of infection of third species with zoonotic pathogens that benefit from common vole population outbreaks. Our study also contributes with unknown (although limited) information on the strains that could be circulating in M. arvalis and show that future typing studies should do it better and grow C. burnetii in pure culture before typing to enhance the chances of characterizing circulating strains. All this information contributed to the state-of-the-art knowledge on C. burnetii ecology and would result in better approaches in the future to predict potential re-emergence hotspots from particular foci.

Lines 174-175: “and it is a unique national-scale molecular survey of the pathogen in wild micromammals”: because the sampling scheme is opportunistic, on large time frame (10 years) and not representative of any micromammals population, the term “unique national scale molecular survey” seems excessive

AR: Unique means non-existing. Currently no wide molecular survey has been performed in the country in micromammals, so even though it is opportunistic, the information is new and useful. We cannot agree in that it does not present representative information of any population of micromammals. We are presenting relevant data for M. arvalis in the areas where demographic outbreaks occur with a higher frequency. We however have smoothed this sentence because we do not pretend to state that this is an outstanding survey since we recognize its limitations.

Lines 177-178: “Major findings reveal (…) the presence of genotypes shared with humans, ticks and domestic animals and reported in different European countries [19,22])”: to remove because only one sample from this study was successfully genotyped

AR: It is true that we have only been able to successfully genotype one sample. However, according to the literature, this genotype has also been detected in people, ticks and domestic animals. We honestly feel that this is an interesting finding of the study.

Table 2: I spotted a mistake regarding the results of Microtus arvalis, when referring to the study from Pluta et al. (Germany, 0/119): there were actually 109 M. arvalis tested (other species were M. Arvicola terrestris (n = 3), Apodemus flavicollis (n = 3), and Apodemus sylvaticus (n = 2).

AR: Thank you for the nuance; we have corrected this in the table.

Lines 196-197: “The wild micromammals surveyed for this study came mostly from areas where livestock is absent or, if present, only present at specific times in the year”: this point clearly requires further details

AR: The focus of the study was not on analyzing the interactions among micromammals and other potential coexisting C. burnetii hosts. We however have added some but limited information in Table 1.

Lines 203-204: “An additional observation supporting the potential reservoir role of M. arvalis was the diagnosis of acute Q fever in one of the researchers”: further details are needed to exclude the possibility of contamination from other species that may have excreted C. burnetii in the environment where the field campaigns took place. Also, details about the conditions of manipulations (use of FFP2 or FFP3 masks, gloves, hood…) are needed? Any event of cutting during the dissections?

AR: As mentioned in the text, this person had no history of exposure to livestock and other wildlife environments and had spent several weeks surveying voles before the onset of the symptoms. The probability of being infected by handling voles was deemed to be higher than that from contaminated aerosols because this was the main activity performed at the infection time. We however cannot rule out any possibility as C. burnetii is almost everywhere.

Lines 214 and beyond: as specified in the major comment section, the discussion about “what it takes to be a reservoir” needs to be more developed and the conclusions of the manuscript adapted accordingly

AR: As commented above, the perception of Wobeser and Haydon and colleagues is different. Wobeser postulates are useful to define the potential relevance of a host species as a reservoir for a pathogen whereas Haydon et al. criteria tend to better define the scenarios at which multi-host pathogens could be maintained as a consequence of the community of interacting hosts with variable roles in pathogen ecology. We have modified this part of the text and avoided any specific mention of any prove of reservoir role, so just discussed on the findings and what these may say.

Reviewer 2 Report

González-Barrio et al. present an interesting study reporting levels of C.burnetti in small mammals in Spain. The sample size is impressive and the results are thoroughly discussed and evaluated. The main (minor) issue I have with the paper is that the description of the studies is very vague, other than short sentence at line 103. It would be useful to know more about the time, date etc that these 16 locations were sampled. I suggest a simple table containing the location, the date the samples were collected, and a brief description of the habitat should be added to the paper. Otherwise it is difficult to interpret the results on figure 1, and Table 1.

Overall the paper is extremely clear and well written, but I have found a small number of minor suggested changes listed below:

  • Line 58: “evolve to more severe and complicated chronic courses” is a bit overly complicated….I would suggest replacing with “develop chronic disease and present with….” or “experience severe infection leading to…”, or something similar.
  • Line 79: “Being C.burnetti a multi-host pathogen….” change to “As C.burnetti is a multi-host pathogen….”
  • Line 90: Suggest changing “along the last four decades” to “over the last four decades”…..
  • Line 116: change “found death” to “found dead”
  • Line 120: suggest removing “an effective”
  • Line 121: change “swab” to “swabs”
  • Line 152: Suggest changing “The rest of micromammal species…” to “The other micromammal species sampled…”
  • Line 161: Suggest changing “A major part of” to “A significant number of”
  • Line 174: I think “National scale” is a slight exaggeration. While your study is quite large it is not a thorough survey of the whole country and it is not spread out over a large enough area to give an accurate national estimate. I suggest changing to “large scale”.

Author Response

Reviewer 2

González-Barrio et al. present an interesting study reporting levels of C.burnetti in small mammals in Spain. The sample size is impressive and the results are thoroughly discussed and evaluated. The main (minor) issue I have with the paper is that the description of the studies is very vague, other than short sentence at line 103. It would be useful to know more about the time, date etc that these 16 locations were sampled. I suggest a simple table containing the location, the date the samples were collected, and a brief description of the habitat should be added to the paper. Otherwise it is difficult to interpret the results on figure 1, and Table 1.

Authors’ response (AR): We greatly acknowledge the time and effort this reviewer put to provide comments that have helped improving the accuracy of the information presented in this study. We considered all reviewer’s comments carefully.

We have included a new table (Table 1) indicating the location, type of vegetation, wildlife or livestock present and cohabiting with the micromammals, as well as whether Coxiella burnetii DNA has been previously detected in any animal species

Overall the paper is extremely clear and well written, but I have found a small number of minor suggested changes listed below:

  • Line 58: “evolve to more severe and complicated chronic courses” is a bit overly complicated….I would suggest replacing with “develop chronic disease and present with….” or “experience severe infection leading to…”, or something similar.
  • AR: This sentence has been modified in the main text.
  • Line 79: “Being C.burnetti a multi-host pathogen….” change to “As C.burnetti is a multi-host pathogen….”
  • AR: This sentence has been changed in the main text.
  • Line 90: Suggest changing “along the last four decades” to “over the last four decades”…..
  • AR: This sentence has been changed in the main text.
  • Line 116: change “found death” to “found dead”
  • AR: Ok, corrected.
  • Line 120: suggest removing “an effective”
  • AR: This sentence has been modified to show why we selected spleen samples as target organ for molecular diagnosis of C. burnetii infection.
  • Line 121: change “swab” to “swabs”
  • AR: Ok, corrected.
  • Line 152: Suggest changing “The rest of micromammal species…” to “The other micromammal species sampled…”
  • AR: It has been changed in the main text.
  • Line 161: Suggest changing “A major part of” to “A significant number of”
  • AR: It has been changed in the main text.
  • Line 174: I think “National scale” is a slight exaggeration. While your study is quite large it is not a thorough survey of the whole country and it is not spread out over a large enough area to give an accurate national estimate. I suggest changing to “large scale”.
  • AR: Ok, we agree with the reviewer. This sentence is not included in the new version of the manuscript.

Reviewer 3 Report

Insights into Coxiella burnetti epidemiology and the role of rodents and small mammals is of high importance. Although there are some language issues in the text the manuscript is well written. The title could be more concise, as “reveal a potential role of micromammals” does not say much. Please make this more concise (or even a bit shorter). I appreciate the effort made in catching all the small mammals, however looking at figure 1 it becomes clear that there is a rather large bias in sample areas and hence in the sample size of the different locations. If this cannot be fixed, it should at least be discussed in the manuscript.

Simple summary: why are the swabs not mentioned in the simple summary?

Material and methods line 109 -> it is stated “some of the individuals captured” -> how many are some? How many were captured in total? How was it decided which once were euthanized?

In the results and discussion more references should be included -> e.g. line 158,159, 169, 181

Also 2 references in the list are not cited in the text -> reference 34 and 35

Looking at the literature some newer articles (2020) could/should be included and discussed in this manuscript as well, as they give more weight to this topic.

The authors mention that animals were sexed and divided into 3 age classes -> however this is not mentioned in the results -> was it looked at, if there were any differences between male/female, young/mature/adult?

Author Response

Reviewer 3

Insights into Coxiella burnetti epidemiology and the role of rodents and small mammals is of high importance. Although there are some language issues in the text the manuscript is well written. The title could be more concise, as “reveal a potential role of micromammals” does not say much. Please make this more concise (or even a bit shorter). I appreciate the effort made in catching all the small mammals, however looking at figure 1 it becomes clear that there is a rather large bias in sample areas and hence in the sample size of the different locations. If this cannot be fixed, it should at least be discussed in the manuscript.

Authors’ response (AR): We are very grateful for the comments, criticisms and suggestions. We have slightly modified the title to make it more concise and to better show the nature of the study. We have clearly discussed the opportunistic nature of our survey so the readers can be aware when interpreting the findings.

Simple summary: why are the swabs not mentioned in the simple summary?

AR: Ok, vaginal swabs have been mentioned in the simple summary.

Material and methods line 109 -> it is stated “some of the individuals captured” -> how many are some? How many were captured in total? How was it decided which once were euthanized?

AR: We cannot show how many of the captured individuals were euthanized since this rate varied along the different studies in which samples were collected, so it is highly variable. We believe that sampling size rather than capture numbers is the relevant parameter in this study, but we want to show how animals were collected. The selection of individuals to be euthanized was random and this is now reflected in the text.

In the results and discussion more references should be included -> e.g. line 158,159, 169, 181

AR: Ok, some references supporting these asseverations have been added. Please, be aware that in previous line 181 we added the references in the summarizing table (Table 3).

Also 2 references in the list are not cited in the text -> reference 34 and 35

AR: Thank you for noticing that gap. Some references are only included in Table 3 and not in the main text.

Looking at the literature some newer articles (2020) could/should be included and discussed in this manuscript as well, as they give more weight to this topic.

AR: Thank you for your suggestion. We’ve updated the list of publications on C. burnetii in micromammals. We have only found new findings in the Czech Republic in 2020 but these are based on serology, so we have not included it in Table 3.

The authors mention that animals were sexed and divided into 3 age classes -> however this is not mentioned in the results -> was it looked at, if there were any differences between male/female, young/mature/adult?

AR: The referee was right. We did indeed register this and further information during the necropsies, but it was out of the scope of this descriptive study analyzing the association of potential risk factors of C. burnetii infection. We have deleted this from the M&M section to avoid confusing the readers.

Reviewer 4 Report

Overall, this is a well researched and written paper. The authors point out limitations to a PCR-only study, without culture and are cautious in tying results to other studies. It would be useful to expand the discussion on genotypes some and explain the limitations for only being able to genotype a single sample.

As the work on Tularemia is cited at length, it would be good to write about the distribution of F.t. and how that may inform (or not) Q Fever - as much work on F.t. in these rodent species occurs in the literature.

While the authors present some overlap in sheep Q-Fever years and rodent cycles, this should be expanded to discuss/describe how each rodents and sheep become infected and if how the sheep may be infected if more rodents are shedding the pathogen. This was not described in the results.

Overall, a nice paper.

Author Response

Overall, this is a well researched and written paper. The authors point out limitations to a PCR-only study, without culture and are cautious in tying results to other studies. It would be useful to expand the discussion on genotypes some and explain the limitations for only being able to genotype a single sample.

AR: Thank you for your thorough revision, the criticisms, comments and suggestions. We have carefully considered them to prepare a new version of the manuscript.

As for the discussion section on the genotype found, more information has been included in the text on the difficulty of being able to genotype wildlife samples.

As the work on Tularemia is cited at length, it would be good to write about the distribution of F.t. and how that may inform (or not) Q Fever - as much work on F.t. in these rodent species occurs in the literature.

AR: We find this suggestion relevant, but honestly feel it difficult to discuss on how infections of micromammals with Francisella tularensis would relate to infections with C. burnetii, especially due to the poor knowledge of the factors that influence C. burnetii dynamics in the environment where M. arvalis and other hosts co-exist. We have added some more information on the distribution of F.t. in Spain.

While the authors present some overlap in sheep Q-Fever years and rodent cycles, this should be expanded to discuss/describe how each rodents and sheep become infected and if how the sheep may be infected if more rodents are shedding the pathogen. This was not described in the results.

Overall, a nice paper.

AR: We also find this potential association interesting, but this needs to be further explored if something else has to be added to the observation. We feel this goes beyond the scope of the study. Our findings are limited to properly discuss on how infections in sheep relate to vole demographic changes. We find it relevant mentioning the potential observed association but feel going beyond is out of the scope and the limits of the available information.